# BLOOD: Bi-level Learning Framework for Out-of-distribution Generalization

## ABSTRACT

Empirical risk minimization (ERM) based machine learning algorithms have suffered from weak generalization performance on the out-of-distribution (OOD) data when the training data are collected from separate environments with unknown spurious correlations. To address this problem, previous works either exploit prior human knowledge for biases in the dataset or apply the two-stage process, which re-weights spuriously correlated samples after they were identified by the biased classifier. However, most of them fail to remove multiple types of spurious correlations that exist in training data. In this paper, we propose a novel bi-level learning framework for OOD generalization, which can effectively remove multiple unknown types of biases without any prior bias information or separate re-training steps of a model. In our bi-level learning framework, we uncover spurious correlations in the inner-loop with shallow model-based predictions and dynamically re-group the data to leverage the group distributionally robust optimization method in the outer-loop, minimizing the worst-case risk across all batches. Our main idea applies the unknown bias discovering process to the group construction method of the group distributionally robust optimization (group DRO) algorithm in a bi-level optimization setting and provides a unified de-biasing framework that can handle multiple types of biases in data. In empirical evaluations on both synthetic and real-world datasets, our framework shows superior OOD performance compared to all other state-of-the-art OOD methods by a large margin. Furthermore, it successfully removes multiple types of biases in the training data groups that most other OOD models fail.

## 1 INTRODUCTION

Conventional machine learning algorithms are relying on the empirical risk minimization (ERM) method when they should learn from given data, and in many application areas, this approach has shown successful performance with high prediction accuracy. However, if a model learns spurious correlations during training, it can often fail with poor generalization performance, which is known as the out-of-distribution (OOD) generalization problem. Furthermore, in recent studies, it has been shown that ERM-based methods more easily learn such unstable correlations in the dataset and result in a poor generalization performance on real-world applications (Beery et al., 2018; Ilyas et al., 2019; Geirhos et al., 2018; de Haan et al., 2019; Koh et al., 2021).

To address this problem and obtain a robust de-biased model, many approaches have been proposed for the cases where biases are known beforehand or not. When biases are known as a priori, some studies applied adversarial training to remove biases from representations (Belinkov et al., 2019a;b) or re-weighting training samples (Schuster et al., 2019), and assembling predictions of a biased model and the base model for ensemble with a product of experts (Hinton, 2002; He et al., 2019; Clark et al., 2019; Mahabadi et al., 2020). However, these works are designed for a specific type of bias and thus require extra domain knowledge to generalize to new tasks. Moreover, without such prior knowledge for biases in the data, they are hard to be applied to practical applications. For the case of having no prior knowledge of spurious correlations, the most popular approach is leveraging the prediction result of a shallow model or weak learner while assuming them as a biased classifier. Since predictions of a biased classifier can provide useful clues for the spurious correlation it has learned, to learn from the weak models' mistakes, they down-weight the potentially biased examples while training a robust model (Mahabadi et al., 2020). Although these works are

more general approaches and save a lot of human efforts for finding biases in established datasets, we found that they still cannot effectively remove multiple types of biases existing in data groups collected from different environments.

Another type of effective OOD generalization approach is group distributionally robust optimization (group DRO) algorithm which alleviates model biases by minimizing the worst-case risk over a set of human-defined training groups (Hu et al., 2018; Sagawa* et al., 2020). In this method, the choice of how to group the training data allows us to introduce a prior knowledge of spurious correlations into optimization. However, finding multiple types of biases and accordingly constructing data groups are laborious processes. Therefore, a simple grouping algorithm is proposed in a recent study (Bao et al., 2021), which splits the training dataset based on the prediction results of biased classifiers. Our approach is also aimed to create data groups that are informative for the multiple underlying biases in the training dataset so that minimizing the worst-case risk over all those data groups can provide a robust classifier. In practical settings where there is little or no prior information about the biases, most de-biasing methods, which automatically identify potential biases in the training data, cannot discover all spurious correlations existing in the dataset. Furthermore, splitting training data into several static data groups cannot effectively represent the effect of multiple biases existing in the dataset.

In this work, first, we propose a novel strategy for discovering bias and splitting training data for a group-based de-biasing algorithm. Based on the prevailing automatic bias identifying approaches (Utama et al., 2020b; Sanh et al., 2021), we train a shallow model for each group in a batch and dynamically re-group the environments according to the prediction correctness of the shallow model over all other environments in the batch. This batch-wise dynamic data re-grouping strategy allows us gradually uncover multiple unknown biases in the dataset while training a model. A shallow model tends to quickly overfit to surface form information, especially when they are trained with a small training data setting (Utama et al., 2020b); therefore, if we re-group the samples in a batch, based on its biased prediction results, we can more effectively account for the various unknown biases in the training dataset. Furthermore, when this approach is combined with the group-based de-biasing method (group DRO), we can train a more robust classifier by minimizing the worst-case risk over all interpolations of those dynamic data partitions.

Second, we also propose a unified end-to-end learning framework for a stable classifier. Our framework is a bi-level learning process which extends the min-max objective of group DRO with the unknown bias discovering and grouping method. In the inner level of optimization, it discovers the biases in the environment of each batch by applying the dynamic data re-grouping method, and these re-partitioned data groups are used for the group DRO algorithm, which minimizes the worst-case risk in the outer level of optimization. We coin our novel learning framework as a Bi-level Learning framework for OOD generalization (BLOOD) and evaluate its OOD performance in both synthetic and real-world environments. In the empirical evaluation, our framework shows 47% percent improvement on the Colored MNIST dataset and achieves the best results in real-world datasets (Camelyon17-wilds, FMoW-wilds) compared to other OOD methods.

The main contributions of our work are the following: (a) we show that dynamically re-grouping the subset of environments, based on the predictions of the shallow model, gradually uncovers multiple types of spurious correlations existing in a dataset; (b) we integrate automatic unknown bias identifying and grouping process to the group DRO by formulating a bi-level optimization objective; (c) we propose a unified end-to-end learning framework which does not need prior knowledge of multiple dataset biases to obtain robust models, but automatically removes various unknown biases from out-of-distribution data.

## 2 METHOD

### 2.1 OVERVIEW

Consider a set of $N$ training environments $\mathcal{E}_{tr} = \{e_i\}_{i=1}^N$ where each environment $e_i$ is composed of input-label pairs $\{(x_k^e, y_k^e)\}_{k=1}^n$. Our main goal is to train a stable classifier from these environments so that it can be generalized to any new test dataset. We do not make any assumption on the biases present (or not) in the dataset and rely on letting the shallow model discover them during training.

For this type of problem, recent de-biasing methods (Utama et al., 2020b; Sanh et al., 2021; Liu et al., 2021; Bao et al., 2021) apply a two-stage process: (a) train a separate biased classifier by learning from spurious correlations in the environments and use the prediction correctness of the biased classifier for identifying a biased sample (Utama et al., 2020b; Liu et al., 2021) or creating new environment partitions for group-based de-biasing methods (Bao et al., 2021). (b) train a robust classifier by re-weighting biased samples or applying product-of-experts (Hinton, 2002; Sanh et al., 2021), confidence regularization (Utama et al., 2020a) to the outcome of the first stage.

In our approach, we integrate these two stages into a single unified learning framework by leveraging a bi-level optimization structure. With the bi-level setting, we can identify unknown biases in the inner-level and remove the discovered biases with group DRO in the outer-level. In the inner-level, to automatically discover unknown biases in the environments, we train a shallow model for each environment in a batch and apply an environment-specific classifier to partition all other environments in the batch, based on its prediction correctness. Since a shallow model is more prone to rely on shallow heuristics, we can obtain a biased classifier after few-shot learning in the batch, and this procedure is iteratively performed while training a model. In the outer-level, the dynamically partitioned environments, which are obtained from inner-level optimization, are provided to the group DRO algorithm and used for minimizing the worst-case risk over those partitions. In the next section, we describe each component of our bi-level learning structure with its corresponding contribution.

## 2.2 Bi-level Learning Framework for De-biasing

Our learning framework consists of two training objectives for bi-level optimization, in which the inner objective is nested within the outer objective. The inner training objective learns spurious correlations in the dataset, which are unknown, and uses them for re-grouping other environments. The outer objective learns only stable correlations by minimizing the worst-case risk over these groups.

### 2.2.1 Inner Objective: Learning Bias to de-bias

The inner-level optimization is aimed to automatically discover unknown spurious correlations, based on the prediction result of a biased classifier (weak learner), and accordingly re-partitions the environments to provide groups to the outer-level process. Therefore, we need to train a biased model which mostly follows spurious correlations in each environment. Most of the other de-biasing methods, exploiting the prediction results of a biased classifier, use a pre-built biased model by training it with the full set or a small subset of training data. In contrast to those works, we do not train a separate biased model; instead, we dynamically obtain a shallow model for each environment in a batch during inner-level optimization. To obtain biased models while training stable classifier, we get insight from the deep neural network's tendency to exploit simple patterns in the early stage of the training, which is also observed in other researches (Arpit et al., 2017; Liu et al., 2020). Since spurious correlations are commonly characterized as simple surface patterns, we expect that models' rapid performance gain is attributed to their reliance on simple surface patterns (Utama et al., 2020a). In the same context, after a model is trained with only a small number of samples from an environment, we expect it to perform as a biased classifier, which achieves high accuracy mostly on the biased examples while still performing poorly on the rest of the samples from other environments.

For each batch learning step in the inner optimization, a shallow model is trained and used for re-grouping as follows:

**Step 1** : For each sampled environment $e_i$ in a batch, temporarily train an environment-specific classifier $f_{\phi_i}$ with few-shot learning.

**Step 2** : For all sampled environments in the batch where $j \neq i$, use the trained classifier $f_{\phi_i}$ to partition each of them into two parts based on the prediction correctness of the classifier.

**Training a shallow model**  At the beginning of a model training, a model $f_\theta$ is randomly initialized with $\theta$. While training a classifier, we obtain a shallow model for each batch training step by applying a few gradient descent steps to the model's parameters $\theta$, so that the model can quickly overfit to

---

**Algorithm 1** BLOOD: Bi-level Learning Framework for OOD Generalization

---

**Input**: Step sizes $\alpha, \beta, \gamma$; A set of training environments $\mathcal{E}_{tr} = \{e_i\}_{i=1}^N$
**Initialize**: $\theta$ and $\boldsymbol{q} = [q_1^\odot, q_1^\otimes, \cdots, q_N^\odot, q_N^\otimes]$

1: **while** not done **do**
2:    **for** all $e_i \in \mathcal{E}_{tr}$ **do**
3:       $\phi_i = \theta - \alpha\nabla_\theta\mathcal{L}_{e_i}(f_\theta)$                  {Inner-loop optimization for a shallow model}
4:       $e_j \sim \mathcal{E}_{tr} \setminus e_i$                                {Sample training environment $e_j$}
5:       $G_{i\to j}^\odot, G_{i\to j}^\otimes = Partition(f_{\phi_i}, e_j)$           {Partition $e_j$ into $G_{i\to j}^\odot$ and $G_{i\to j}^\otimes$}
6:       $q_j^\odot \leftarrow q_j^\odot \exp(\gamma\mathcal{L}_{G_{i\to j}^\odot}(f_{\phi_i})), q_i^\otimes \leftarrow q_i^\otimes \exp(\gamma\mathcal{L}_{G_{i\to j}^\otimes}(f_{\phi_i}))$     {Update group weights}
7:    **end for**
8:    $\boldsymbol{q} \leftarrow \boldsymbol{q}/\sum_i(q_i^\odot + q_i^\otimes)$                          {Normalize group weights $\boldsymbol{q}$}
9:    $\theta \leftarrow \theta - \beta\,\nabla_\theta \sum_i \sum_j q_j^\odot\mathcal{L}_{G_{i\to j}^\odot}(f_{\phi_i|\theta}) + q_j^\otimes\mathcal{L}_{G_{i\to j}^\otimes}(f_{\phi_i|\theta})$    {Outer-loop optimization}
10: **end while**

---

surface form information while $f_\theta$ is still conditioned on the previously learned $\theta$. This temporarily trained shallow model parameter $\phi_i$, which are optimized for surface patterns of $e_i$, is updated by following stochastic gradient descent (SGD) step:

$$\phi_i = \theta - \alpha\nabla_\theta\mathcal{L}_{e_i}(f_\theta) \tag{1}$$

where $\alpha$ is a step size for the inner optimization, and $\mathcal{L}_{e_i}(f_\theta)$ is an expected loss of $f_\theta$ over the data sampled from $e_i$. We show only a single gradient update procedure in Equation 1 for the notational simplicity.

**Batch-wise dynamic data re-grouping**    After a biased classifier $f_{\phi_i}$ is trained for the environment $e_i$, we dynamically re-partition all other training environments in the batch. The $f_{\phi_i}$ is applied to all other environments $e_j$ in the batch, where $j \neq i$, and according to its prediction correctness, samples in each environment $e_j$ are re-grouped into two parts, a correctly predicted sample group $G_{i\to j}^\odot$ and a incorrectly predicted sample group $G_{i\to j}^\otimes$. These dynamically partitioned groups are then used for training a stable classifier by minimizing the worst-case risk over these groups in the outer-level.

As a similar group-based de-biasing approach, Predict then Interpolate (PI) (Bao et al., 2021) also uses a biased classifier for re-partitioning other environments. Its main difference compared to our method is that they fully train a separate model with the entire dataset and use it for partitioning all environments into static groups. Once their data groups are statically assigned, it is fixed throughout the optimization process of group DRO. Although PI has shown its theoretical and empirical effectiveness as a de-biasing method, its static grouping strategy, which relies on a fully-trained biased classifier, constrains it from discovering multiple types of spurious correlations in the environments.

In contrast, our framework uses a shallow model, which is trained with a small subset of each environment in the batch. Since we are using a different biased classifier per batch, we are continuously learning a bias and producing new data partitions for each batch, representing our dynamic data re-grouping strategy. This dynamic re-grouping strategy provides various group combinations for the outer-level optimization process, so that it can effectively remove various types of biases in the environments.

Furthermore, since our shallow model has overfitted to the surface patterns in a small subset of data, it is more likely to capture multiple types of bias in the environments, as shown in other research (Utama et al., 2020a). This property can be a disadvantage for a de-biasing method, which applies re-weighting to the biased examples, because it reduces the effective training data size for a model. However, in our approach, it enables us to uncover multiple types of biases in the dataset. Because, multiple types of biases can be considered while minimizing the worst-case risk over various types of group configurations.

### 2.2.2 Outer Objective: Minimizing Worst-Case Risk Over Data Groups

The goal of the outer-level objective is to remove spurious correlations that have been identified during the inner-level optimization. For this purpose, we apply the group DRO algorithm in the

outer-level, to iteratively minimize the worst-case risk over dynamically re-partitioned groups. Although the baseline group DRO algorithm is an effective group-based de-biasing method, it still requires data groups to be explicitly defined by the prior knowledge on biases in the training data. In our framework, we already automatically identified unknown biases in the environments and dynamically re-grouped environments in the inner-loop. Therefore, we can directly provide them to the group DRO during the outer-level optimization.

In our bi-level optimization setting, we integrate our dynamic re-grouping scheme with online group DRO (Sagawa* et al., 2020) that minimizes the worst-case of convex combinations of the group risks. The objective of our framework is formulated as following min-max problem:

$$\min_{\theta} \max_{\boldsymbol{q}} \sum_{i} \sum_{j \neq i} q_j^{\odot} \mathcal{L}_{G_{i \to j}^{\odot}}(f_{\phi_i|\theta}) + q_j^{\otimes} \mathcal{L}_{G_{i \to j}^{\otimes}}(f_{\phi_i|\theta}) \tag{2}$$

where $\boldsymbol{q} = [q_1^{\odot}, q_1^{\otimes}, \cdots, q_N^{\odot}, q_N^{\otimes}]$ denotes a coefficients vector for convex combination, and $q_i^{\odot}$ and $q_i^{\otimes}$ denotes group weights for correctly predicted sample group $G_{i \to j}^{\odot}$ and incorrectly predicted sample group $G_{i \to j}^{\otimes}$, respectively. To solve this min-max problem, we interleave gradient-based iterative updates on $\theta$ and $q$; update $q$ with exponentiated gradient ascent, so that groups with high-risk get high weights, and model parameters $\theta$ are updated with SGD.

While training a robust classifier with our bi-level learning framework, the parameters $\theta$ are optimized by the group-wise losses obtained from shallow models in the inner-level. In the early stages of learning, a shallow model can learn a bias after few-shot learning, then the loss difference between two groups, $G_{i \to j}^{\odot}$ and $G_{i \to j}^{\otimes}$, becomes larger, and optimizing $\theta$ over the worst-case group risks can effectively prevent the model $f_\theta$ from learning such biases.

We formulate a bi-level optimization problem by integrating Equation 1 and optimization objective 2 to the inner and outer optimization objectives of the bi-level learning setting. Our framework iteratively uncovers biases in the inner-loop and de-biases them in the outer-loop while optimizing to obtain a robust classifier. The full procedure of our learning framework is described in Algorithm 1.

## 3    EXPERIMENTS

We evaluate our framework on both synthetic and real-world tasks and also compare the OOD performance with other state-of-the-art algorithms, such as ERM, Invariant Risk Minimization (IRM) (Arjovsky et al., 2019), Group DRO, and PI. IRM is an optimization framework based on the theory of Invariant Causal Prediction (Peters et al., 2016), which learns representations that are simultaneously optimal across all environments. For training a model with PI, we train environment-specific classifiers for each of all training environments and randomly pick the environment for evaluation.

### 3.1    SYNTHETIC TASKS: COLORED MNIST

In this section, we demonstrate the multiple biases identification performance of our framework on a synthetic task, Colored MNIST dataset. While following Arjovsky et al. (2019) as a baseline, we also inject more spurious correlations to the original Colored MNIST. For color attributes, green or red are applied to have strong spurious correlations with the class labels. We design these correlations to vary across the environments so that the model which exploits the unstable color attributes cannot guarantee the OOD generalization performance. In detail, we consider two training environments with coloring probability $p_e = 0.1$ and $p_e = 0.2$, respectively, and this indicates that the color attribute is correlated with label as $1 - p_e$. We add noise with probability $p_c = 0.25$ to

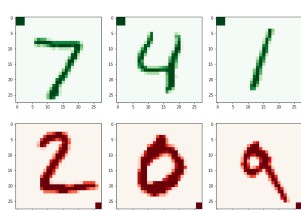

Figure 1: Image samples of Colored MNIST task with additional patch feature.

the shape attribute so that the shape attribute is less correlated with the target label, making the correlation only 75%.

Table 1: Test accuracy (%) of different algorithms on the Colored MNIST task in 5 trials (mean $\pm$ standard deviation). $\delta_{gap}$ indicates the generalization gap of the model between i.i.d and OOD test environments. Note that the highest test accuracy for the i.i.d test environment does not guarantee high performance on the OOD.

| Algorithm | Bias: Color | | | Bias: Color & Patch | | |
|---|---|---|---|---|---|---|
| | Test (i.i.d) $p_e = 0.1$ | Test (OOD) $p_e = 0.9$ | $\delta_{gap}$ | Test (i.i.d) $p_e = 0.1$ | Test (OOD) $p_e = 0.9$ | $\delta_{gap}$ |
| ERM | $88.6 \pm 0.3$ | $16.4 \pm 0.8$ | -72.2 | $93.7 \pm 0.3$ | $14.0 \pm 0.5$ | -79.7 |
| IRM | $71.4 \pm 0.9$ | $66.9 \pm 2.5$ | -4.5 | $93.5 \pm 0.2$ | $13.4 \pm 0.3$ | -80.1 |
| Group DRO | $89.2 \pm 0.9$ | $13.6 \pm 3.8$ | -75.6 | $92.3 \pm 0.3$ | $14.1 \pm 0.8$ | -78.2 |
| PI | $70.3 \pm 0.3$ | $70.2 \pm 0.9$ | **-0.1** | $85.4 \pm 0.9$ | $15.3 \pm 2.7$ | -70.1 |
| BLOOD (Ours) | $70.5 \pm 1.1$ | $\mathbf{70.7 \pm 1.4}$ | **0.2** | $68.3 \pm 2.3$ | $\mathbf{62.3 \pm 3.3}$ | **-6.0** |
| Optimal | 75 | 75 | 0 | 75 | 75 | 0 |

Moreover, we inject additional spurious correlation by creating small patches of noise to the corner of the image so that the locations of these patches are strongly correlated with the labels. Similar to the coloring MNIST digits, we design the patch attributes to have unstable correlations across the training environments with the probability $p_e$. We add $(3 \times 3)$ patch in the top left corner or the bottom right corner of the image, depending on the labels. The new patch features are independent of other types of spurious correlations, but they have strong but unstable correlations with the target labels. Figure 1 shows the image samples of our Colored MNIST task with additional patch bias.

For algorithm evaluation, we consider both independent and identically distributed (i.i.d) data and the OOD test environments with $p_e = 0.9$ where the correlation between color and label is reversed compared with training environments. The model that exploits color attributes for the prediction can achieve high accuracy in the i.i.d test environment, but it will fail in the OOD test. The purpose of the Colored MNIST task is to classify digits solely based on the shape of digits without relying on other spurious features, such as colors, to achieve good OOD generalization.

Table 1 shows evaluation results on the Colored MNIST task according to the types of injected spurious correlations. In all cases, BLOOD achieves state-of-the-art OOD performance on Colored MNIST. For the case with multiple types of biases (*Color & Patch*), BLOOD outperforms PI (Bao et al., 2021) by 47%, which clearly demonstrate that BLOOD can more effectively de-bias multiple types of spurious correlations in the dataset. Moreover, from the results, BLOOD consistently shows superior generalization performance with low variance for both i.i.d and OOD test data.

Figure 2 shows the changes in the prediction accuracy on test environments while varying noise probability for the shape and other biased attributes to the labels. Our framework shows robust prediction performance against the change of various spurious correlations, which is closer to the oracle pattern. In contrast, PI shows prediction patterns more relying on spurious correlations.

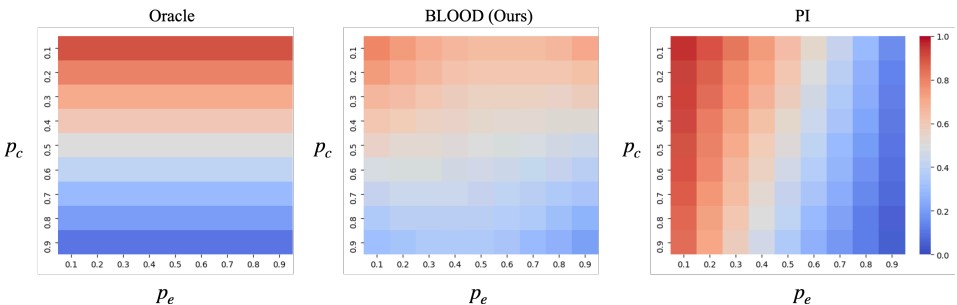

Figure 2: Visualization of various test accuracy on Colored MNIST with varying noise probability for shape $p_c$ and for color and patch $p_e$. Our BLOOD shows robustness performance against the change of $p_e$, while PI highly depends on the spurious correlations.

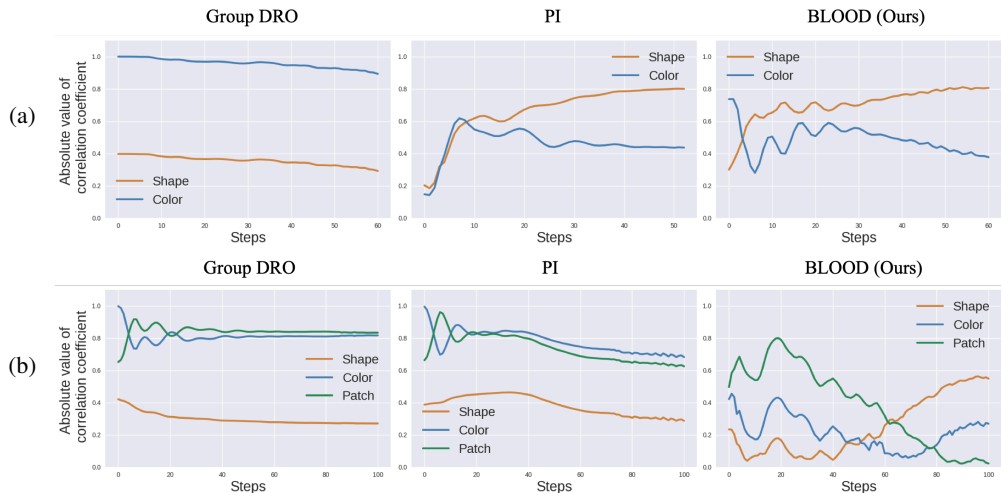

Figure 3: The Pearson correlation coefficients between model's predictions and bias features on the Colored MNIST task with (a) a single spurious feature *Color* and (b) two types of spurious features, *Color* and *Patch*. The prediction results of BLOOD show high correlations with the shape of digits for both (a) and (b) cases, while PI fails to de-bias when there are two types of spurious correlations in the data.

We also analyze how much each attribute affects the prediction results of a model. In Figure 3, for each OOD method, the correlation between the predictions of a model and each attribute (*Shape*, *Color*, *Patch*) are shown according to the training steps, on the environment with $p_e = 0.9$. When there is only *Color* bias, both PI and BLOOD successfully de-bias a model by reducing the correlation between *Color* and the label. However, if an extra *Patch* attribute is added as spurious correlation, only BLOOD reduces the effect of both *Color* and *Patch* attributes on model prediction and recover correct *Shape* correlation. All other OOD methods, group DRO and PI, suffer from biases and fail to discover true correlations across environments.

To verify the effectiveness of batch-wise bias identification and dynamic re-grouping method of BLOOD, in Figure 4, the grouped result of PI and BLOOD with each group's correlations to the bias attributes (*Color*, *Patch*) are visualized for the Colored MNIST task. Compared to PI, which relies on pre-built biased classifiers, BLOOD use a shallow model trained with samples of each environment in a batch; therefore, it dynamically produces new data partitions for each training step, as shown in Figure 4b. Since grouping depends on the bias identification performance of the classifier, each pair of groups(correct & incorrect) predicted by a learned bias, Pearson correlation to the bias attribute should be positive for the correct group, $G_{1\to2}^{\odot}$, and negative for the incorrect group, $G_{1\to2}^{\otimes}$, for example, if correct one is 0.99 then the other is -0.97 (Bao et al., 2021).

As shown in Figure 4b, in the early step of training, the shallow model of BLOOD learns *Color* bias and splits the other environment, so that correlation of the correct group is 0.99 and the other is -0.97. With these correlations, *Color* bias can be easily addressed with the following group DRO algorithm in the outer-level. When there are multiple types of biases, BLOOD learns each of them in

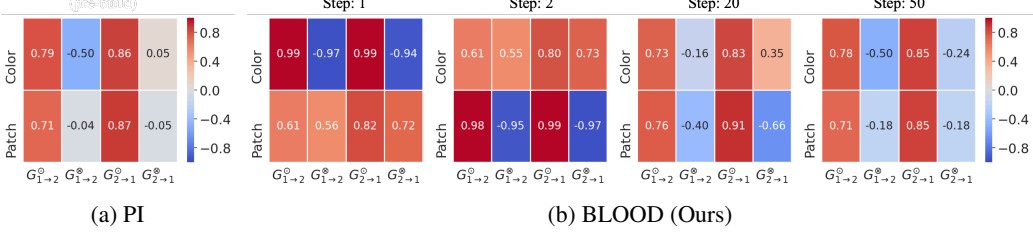

Figure 4: Visualization of created groups with Pearson correlation coefficient between the label and bias features on Colored MNIST task. Our BLOOD dynamically re-groups the training data at each learning step while PI constructs single static groups.

an ordered way. The grouping of BLOOD at step 1 shows that *Color* bias is identified by a shallow model, and environments are partitioned accordingly. At step 2, we can also verify that BLOOD has learned *Patch* bias, and it re-groups the environments with 0.98, -0.95 *Patch* correlations. Therefore, it is shown that BLOOD gradually discovers multiple types of biases in the dataset by adopting the batch-wise shallow model as a biased classifier. However, for the case of the PI in Figure 4a, the training environments are re-grouped only once by a fully-trained environment-specific model. This biased classifier learns multiple biases together, therefore, its data partition should represent the overall effect of multiple biases. In this setting, interpolating these static partitions may not be enough to approximate oracle distribution. Because it does not have a chance to gradually refine the defined groups to account for multiple types of biases. In this experiment, PI fails to remove two biases in the dataset, as shown in Figure 3.

## 3.2 REAL-WORLD TASKS

### 3.2.1 CAMELYON17-WILDS

In the field of machine learning-based medical image processing, the OOD generalization is a critical problem to obtain a universally applicable prediction model in several hospitals. Camelyon17-wilds dataset (Bandi et al., 2018; Koh et al., 2021) is a medical image classification benchmark that explicitly targets the OOD generalization problem. The main goal of Camelyon17-wilds is to achieve high prediction accuracy for predicting the presence of tumor tissue on image patches taken from hospitals not included in the training data.

The training data consists of image patches from three hospitals, and the test data contains patches from a hospital that does not exist in training data. Also, the hospital for the test data provides the most visually unique patches among the data from other hospitals. The final model selection is performed based on the test accuracy on OOD validation data that has different distribution from both the training and test data. The patches for validation data are also taken from a different distribution with the training data but have more similar visual patterns with training data than the test data.

Table 2 shows the experimental results on the Camelyon17-wilds dataset. Although the ERM shows the best train accuracy, it degrades on the OOD test data. From the results, BLOOD achieves the state-of-the-art OOD generalization performance and also outperforms all other algorithms by a large margin.

Table 2: Train and OOD test accuracy (%) on the Camelyon17-wilds in 3 trials (mean $\pm$ standard deviation). We consider average accuracy on the OOD test environment.

|              | ERM          | IRM          | Group DRO    | PI           | BLOOD (Ours)     |
| ------------ | ------------ | ------------ | ------------ | ------------ | ---------------- |
| Train acc    | $97.3 \pm 0.1$ | $97.1 \pm 0.1$ | $96.5 \pm 1.4$ | $93.2 \pm 0.2$ | $93.0 \pm 1.8$     |
| OOD Test acc | $66.5 \pm 4.2$ | $59.4 \pm 3.7$ | $70.2 \pm 7.3$ | $71.7 \pm 7.5$ | $\mathbf{74.9 \pm 5.0}$ |

### 3.2.2 FMOW-WILDS

FMoW-wilds dataset (Christie et al., 2018; Koh et al., 2021) is another benchmark for OOD generalization, including satellite images taken from various locations and times. The dataset consists of RGB satellite images and labels for the task of classifying 62 different functional purposes of buildings and land, based on given images and metadata, which provide the location and time information of each image. The training data includes the times of images were taken, from 2002 to 2013, while the validation and test data include data collected from 2013 to 2015 and 2016 to 2017, respectively. FMoW-wilds aims to evaluate whether a model can be generalized to the images which will be obtained in the future.

There are five geographic regions in each data split where the images were taken. The OOD generalization performance of each algorithm is evaluated with the worst-region accuracy for the test data where the data collection period does not overlap with the training data. Table 3 shows our model evaluation results on the FMoW-wilds task. From the results, BLOOD achieves the highest worst-region accuracy, outperforming all other OOD methods.

Table 3: The worst-region accuracy (%) for the OOD test environment on the FMoW-wilds in 3 trials (mean $\pm$ standard deviation).

|  | ERM | IRM | Group DRO | PI | BLOOD (Ours) |
|---|---|---|---|---|---|
| Wosrt-region acc | $31.3 \pm 0.17$ | $32.8 \pm 2.1$ | $31.0 \pm 1.6$ | $31.2 \pm 0.3$ | $\mathbf{34.1 \pm 2.5}$ |

## 4 RELATED WORK

**De-biasing with explicit supervision.** Biases are always present as a part of the real dataset, and they degrade a learning model's prediction performance when it is applied to practical applications. To mitigate such biases, some researches de-bias a model with adversarial training (Belinkov et al., 2019a;b), data sample re-weighting (Schuster et al., 2019), or ensemble of biased models (He et al., 2019; Clark et al., 2019; Mahabadi et al., 2020), when biases are known beforehand by the human expert knowledge. However, these methods are not easily applicable to practical applications because prior knowledge for biases in the large-scale dataset is generally inaccessible. Compared to these works, our framework can identify and remove unknown biases without any human annotated bias information.

**De-biasing by a biased classifier.** To automatically identify unknown biases without domain expert knowledge, recent studies utilize a weak learner or shallow model, which is a biased classifier, for verifying whether the data sample is biased. Based on the predictions of the shallow model, they consider re-weighting data samples (Liu et al., 2021), merging the predictions of the biased model and main model with the product of experts (Sanh et al., 2021), or grouping training data based on the prediction correctness of biased models (Bao et al., 2021). Most approaches pre-build a fully-trained biased model to discover biases; however, our learning framework dynamically trains a shallow model at every learning step to uncover multiple types of biases.

**De-biasing with invariant representation.** Since the spurious correlations vary across environments, several researchers focus on finding stable correlations which are invariant over training environments (Peters et al., 2016; Arjovsky et al., 2019; Ahuja et al., 2020; Chang et al., 2020; Lu et al., 2021). IRM (Arjovsky et al., 2019), a symbolic learning paradigm based on this idea, assumes that the model can learn invariant correlations if the classifier on top of the feature embedder is simultaneously optimal for all training environments. However, a recent study demonstrates the potential degeneration of IRM in real-world scenarios (Rosenfeld et al., 2020). Compared to the IRM, our BLOOD gradually removes various spurious correlations so that only stable correlations can remain across all environments.

## 5 CONCLUSION

In this paper, we propose a novel bi-level learning framework for OOD generalization, which is robust to multiple types of dataset biases. Our framework automatically identifies unknown biases during the training process and gradually removes multiple types of biases without any prior knowledge for biases in the dataset. To uncover unknown biases, we presented a novel strategy of training a shallow model for batch-wise bias identification. By dynamically re-grouping the environments in the batch, we effectively de-biased a model, based on the prediction correctness of a shallow model, by leveraging the group DRO algorithm. In our empirical evaluations, BLOOD achieves the state-of-the-art OOD generalization performance on both real-world applications and synthetic tasks. Also, our extensive analyses on the Colored MNIST task show the efficiency of our approach for de-biasing multiple types of spurious correlations. As Future works, we are going to extend BLOOD for the other types of tasks beyond classification.

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

# A EXPERIMENTAL DETAILS

## A.1 COLORED MNIST

We consider two training environments that contain 25,000 MNIST images and one validation environment with 10,000 images. For the test environment, we use official test images of the MNIST dataset with 10,000 data samples. The color and patch attributes are intended to have high correlations with target labels, but these correlations are unstable as they vary across the training environments.

We train a simple MLP with one hidden layer and ReLU activation function. We use a hidden size of 390 and learning rates of $\alpha =$1e$-$1 and $\beta =$1e$-$3 for inner optimization and outer optimization, respectively. We consider three steps of gradient descent in the inner optimization in our experiments. In experiments, we use an early stopping on validation environment with $p_e = 0.2$ that has similar distribution to the training distribution.

## A.2 CAMELYON17-WILDS

The Camelyon17-wilds is a binary classification task whether the central region of a given $(96 \times 96)$ image patch contains any tumor tissue. The dataset contains training data with 302,436 histopathological image patches, OOD validation data with 34,904 patches, and OOD test data with 85,054 patches. Each data split does not overlap with the hospitals where the data was collected.

We use DenseNet-121 (Huang et al., 2017) without pre-trained parameters, training from scratch on the Camelyon17-wilds dataset as the official setting from Koh et al. (2021). We found that the official learning rate 1e$-$3 is too large in our case; thus, we use learning rates of 1e$-$5 for both inner optimization and outer optimization and $L_2$-regularization term of 1e$-$2. We consider only one step of inner gradient descent to reduce the computational overhead.

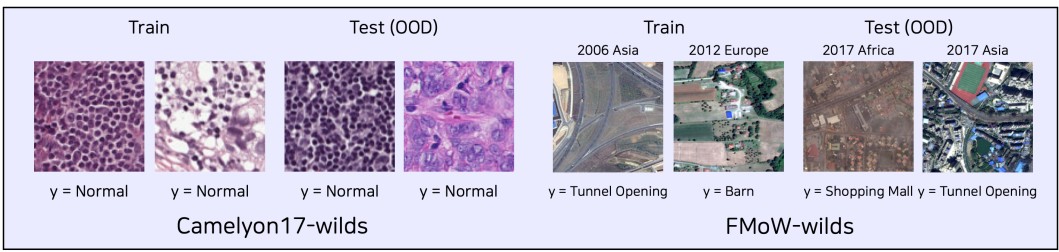

Figure 5: Data examples of Camelyon17-wilds and FMoW-wilds.

## A.3 FMOW-WILDS

The FMoW-wilds is an image classification task that targets to predict 62 categories of building or land use from given $(224 \times 224)$ images. The training split includes 76,863 images taken from 2002 to 2013, and the validation split contains 19,915 images from 2016 to 2018. The OOD test data comprises 22,108 images from the years from 2016 to 2018. All data splits contain images from five regions, Africa, Americas, Oceania, Asia, and Europe. We evaluate the model by measuring the worst-region accuracy on the OOD test data.

We train DenseNet-121 pre-trained on the ImageNet dataset on the FMoW-wilds images. We follow an official hyperparameter from (Koh et al., 2021), optimizing with learning rates of 1e$-$4 for inner and outer learning rate and without $L_2$-regularization. We consider only one step of gradient descent in the inner optimization.

## B    ADDITIONAL EXPERIMENTAL RESULTS

### B.1    SHALLOW MODEL ANALYSIS ON COLORED MNIST

We analyze the predictions of the shallow models and show our shallow models can keep discover biases. Figure 6 shows the incorrect prediction results of shallow models as stacked bar charts. The legend of each graph indicates the class index and attribute indices, which shows the class indices highly correlated with the attributes. For example, the legend ($y = 0$, $Color = 1$, $Patch = 0$) means samples whose color attribute is highly correlated with the label $y = 1$, and patch attribute is highly correlated with $y = 0$.

At the first learning step (step 0) in Figure 6a, most of misclassified samples, predicted as $\hat{y} = 1$, have the *Color* indices as 1, indicating that the shallow models exploit the *Color* for predictions. The shallow models begin to leverage the *Patch* after a few learning steps. The stacked bar charts demonstrate that the shallow models discover biases dynamically and a few steps of gradient descent in the inner optimization can make the robust parameters $\theta$ become parameters for identifying biases.

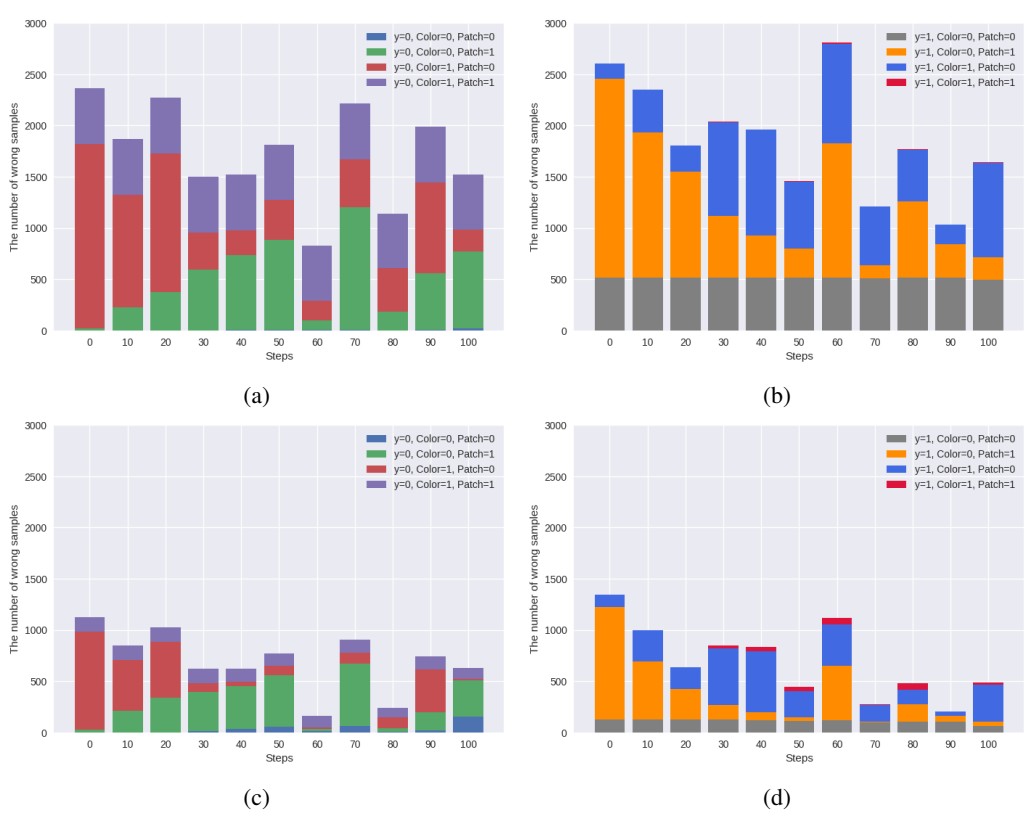

Figure 6: Visualization of the number of incorrectly predicted samples by shallow models $f_{\phi_1}$ and $f_{\phi_2}$ on the Colored MNIST. The stacked bar chart (a) and (b) indicates the misclassified examples of environment $e_2$ predicted by $f_{\phi_1}$, and (c) and (d) indicates incorrect prediction of $f_{\phi_2}$ on environment $e_1$.

Figure 7 includes (a) Pearson correlation between prediction and each attribute, (b) shallow model's confidence for incorrect predictions, and (c) visualization of dominant spurious features for re-grouping process. The confidence is the prediction probability of the shallow models. At the beginning of the learning steps, the confidence is high, indicating high loss for misclassified samples, while it decreases gradually. Considering the confidence for misclassified samples of shallow models with Figure 6, the shallow models still discover biases from the inner optimization conditioned on the parameters $\theta$ at the end of the learning steps while the confidence is low.

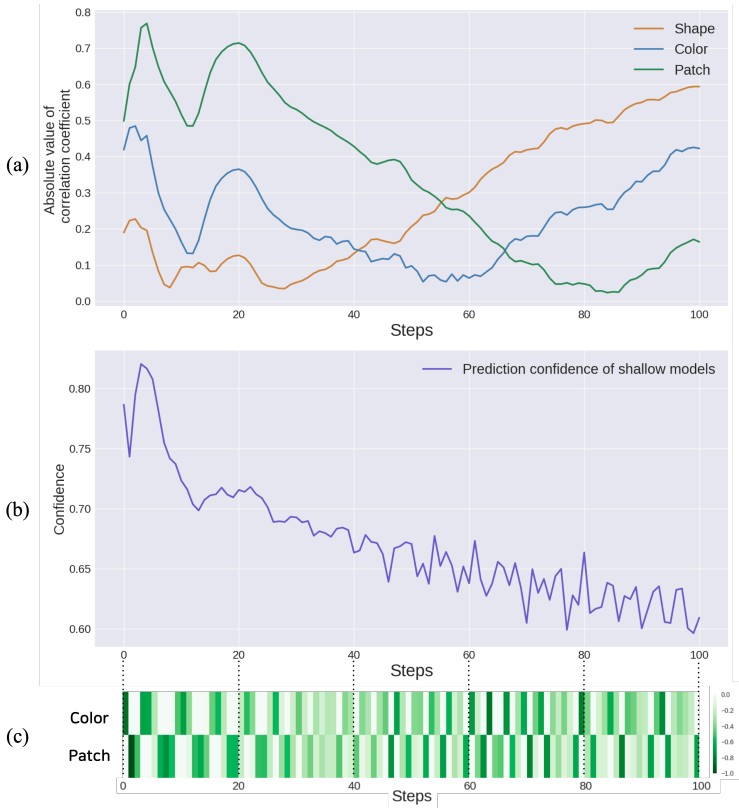

Figure 7: Visualization of (a) Pearson correlation coefficient between spurious features and models prediction, (b) confidence scores of the shallow model's wrong predictions, and (c) dominant spurious attributes for re-grouping. Confidence is measured as the average of the top 20% softmax probabilities values. In (c), darker green indicates strong negative Pearson correlations between attributes and target labels, representing the more dominant features in the crafted groups.

## B.2    CELEBA RESULTS

We evaluate BLOOD on the CelebA dataset, which provides rich information of attributes of each image with 40 binary features. We split training data into two training environments: *Male* and *Female*, and the task is classifying whether input images have blonde hair or not. In table 4, we measure the Pearson correlation coefficient between each attribute and the targets across the environments, *Male* and *Female*, and report three attributes *Black_Hair*, *Brown_Hair*, and *Bushy_Eyebrows* which are highly correlated with labels, and one attribute, *Attractive*, which has relatively low correlation coefficients.

Table 5 shows the test results on the CelebA task. We divide test data into four groups by using each binary attribute and the target, and we measure the worst-group accuracy and average accuracy across those groups. As Sagawa* et al. (2020) and Bao et al. (2021) reported, a model trained in

Table 4: Pearson correlation coefficient of the environment *Male* and *Female* between each attribute and the label across four attributes.

| Attribute | Male | Female |
|---|---|---|
| Black_Hair | $-0.0929$ | $-0.2814$ |
| Brown_Hair | $-0.0264$ | $-0.2691$ |
| Bushy_Eyebrows | $-0.0597$ | $-0.1252$ |
| Attractive | $0.0106$ | $0.0499$ |

Table 5: Test accuracy across four attributes on CelebA dataset.

| | ERM | | DRO | | PI | | BLOOD (Ours) | |
|---|---|---|---|---|---|---|---|---|
| | Worst | Avg | Worst | Avg | Worst | Avg | Worst | Avg |
| Attractive | 67.2 | 85.8 | **90.8** | 92.2 | 90.0 | 91.9 | 89.9 | 92.0 |
| Black_Hair | 76.0 | 90.9 | **89.6** | 93.8 | 88.1 | 93.3 | 88.2 | 92.0 |
| Brown_Hair | 43.7 | 79.2 | 64.4 | 85.7 | 59.8 | 83.8 | **74.7** | 92.0 |
| Bushy_Eyebrows | 72.7 | 86.5 | 72.7 | 88.8 | 81.8 | 90.8 | **90.6** | 92.0 |
| Average | 60.1 | 83.6 | 84.3 | 90.8 | 87.0 | 91.4 | **87.1** | **91.7** |

Table 6: Test accuracy of BLOOD for four different groups on the CelebA task. N/A indicates that there is no test data sample in the group.

| Attribute | $y = 0, a = 0$ | $y = 0, a = 1$ | $y = 1, a = 0$ | $y = 1, a = 1$ | Worst |
|---|---|---|---|---|---|
| Attractive | 91.65 | 92.08 | 89.90 | 94.48 | 89.90 |
| Black_Hair | 88.16 | 99.96 | 92.89 | N/A | 88.16 |
| Brown_Hair | 92.25 | 90.31 | 93.51 | 74.71 | 74.71 |
| Bushy_Eyebrows | 90.63 | 98.91 | 92.91 | 90.91 | 90.63 |

ERM learns spurious associations; thus, it is vulnerable to group shifts, showing degraded worst-group accuracy than average accuracy for each attribute. Among the attributes, the performance degradation of ERM is the most severe for *Brown_Hair*, and our algorithm achieves the highest worst-group accuracy than any other baselines. Table 6 provides the detailed results of our algorithm on the CelebA task.

We analyze created groups by shallow models to verify their ability to discover biases. As shown in Table 5, we can naively assume that the attribute *Brown_Hair* is a biased feature since the generalization performance for the test environment is significantly degenerated. Figure 8 shows Pearson correlation coefficient between targets and attributes of created groups with incorrectly predicted samples, (a) $G^{\otimes}_{Female \rightarrow Male}$ and (b) $G^{\otimes}_{Male \rightarrow Female}$.

The results in Figure 8 demonstrate that the shallow models successfully discover *Brown_Hair* even though *Black_Hair* is the most correlated attribute with targets. Moreover, the groups are dynamically constructed for every learning steps while the effect of bias feature on shallow models decreases as learning progresses.

We also analyze the shallow model's confidence on the CelebA dataset in Figure 9. As the analysis of confidence of shallow model on Colored MNIST in Figure 7, the confidence decreases as learning progresses. In effect, the shallow models keep learning biases with few-show learning, their confidence scores are gradually decreased as the underlying robust model becomes more robust.

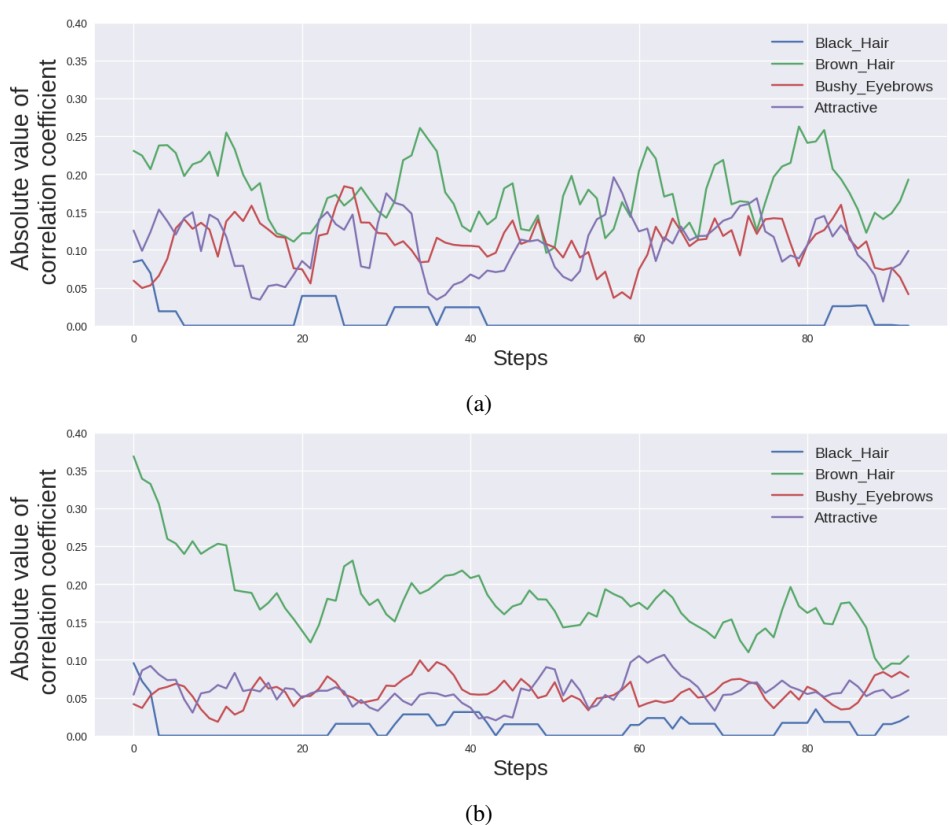

(a)

(b)

Figure 8: Pearson correlation coefficients between the target labels and attribute values of created misclassified groups, (a) regrouping the environment *Male* by shallow model trained on *Female*, $G^{\otimes}_{Female \to Male}$ and (b) $G^{\otimes}_{Male \to Female}$.

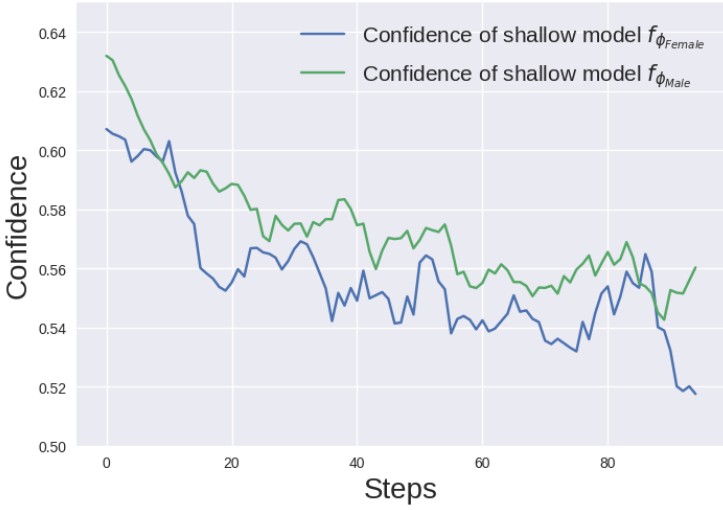

Figure 9: Confidence scores of the shallow model's wrong predictions on disjoint environment. Confidence is measured as the average of the top 20% softmax probabilities values.

