# OpenReview forum: "BLOOD: Bi-level Learning Framework for Out-of-distribution Generalization"
_ICLR.cc/2022/Conference — ICLR 2022 Submitted_

### Official Review · Reviewer_9XTF · 2021-10-29

**Correctness:** 3
**Technical Novelty And Significance:** 1
**Empirical Novelty And Significance:** 2
**Recommendation:** 5
**Confidence:** 4

**Main Review:**

Strengths:

The procedure proposed in this paper is a natural choice, the implementation is easy and can be incorporated in many off-the-shelf machine learning training algorithms.  In particular, given the recent rapid development of bi-level programming, their proposal might receive more refinement or improvement. The performance of the proposal is evaluated in synthetic datasets and shows substantial advantages when more than one spurious feature is present in the data, compared to other existing approaches, including IRM and PI. Thus given the failure of IRM proposed by Arjovsky et al., the proposed method of this paper might be a promising direction to try and see if the beautiful IRM concept can actually become a useful thing in practice.

Weaknesses:

Apparently, this is an empirical paper: proposing a method, showing that it does what the authors expect in synthetic experiments and that it has practical relevance in real data. So I would not pick on the fact that there are no theoretical guarantees at all.

The authors demonstrate a scenario under which their adaptive method outperforms other existing methods like IRM and PI. The authors explained clearly why PI underperforms when there are two types of spuriously correlated features in their synthetic experiments. But for an empirical paper, I would love to see more discussions on when their methods might fail, which could give readers more insights on what is the best application scenario of the proposed method. I think this is the major limitation of this work.

The method, at least based on the current form, is based on shallow learning for identifying spurious correlations. For instance, the authors claimed, "... spurious correlations are commonly characterized as simple surface patterns". I believe the authors need to provide more evidence or explanation to support such a claim and evaluate their method when such a premise does not hold.

For the real data application part, I understand the authors' method achieves the highest OOD accuracy. But I think this section seriously lacks necessary explanations and details. For the two datasets, what are the potential source of biases? Just as in the synthetic experiments, the authors can easily use what BLOOD learned and see if there is any feature whose correlation with the prediction accuracy decreases as the algorithm updates, and also check if those features make any sense. Without such a sanity check, it is hard for me to trust BLOOD in such realistic settings yet, in particular considering that the improvement over some other methods is not that impressive (e.g. 2-3 %).

Minor language issues:

I understand ICLR's policy which asks referees not to pick on language issues. But the current paper contains numerous typos, grammatical errors, and confusing sentences. Just to name a few:
(1) The second paragraph of Section 2.1 is difficult to read because of the incorrect grammar;
(2) Several "can not" should be "cannot";
(3) The first paragraph of Section 2.1.1 "To obtain biased models while a training stable classifier" should be "To obtain biased models while training stable classifier";
(4) Section 2.1.1, in step 2 of the algorithm, "use a trained classifier" should be "use the trained classifier";
(5) Before Figure 2, "which is more close to" should be "which is closer to".
I cannot list all of them. But the authors should seriously take a more careful proofread of the paper if their paper receives more positive feedback from the referees.

**Summary Of The Paper:**

This paper develops a bi-level algorithm that aims at automating the process of identifying biases in data and performing data re-weighting and splitting to eventually train a de-biased model. Numerical experiments are provided to demonstrate the performance of their approach.

**Summary Of The Review:**

Strengths: A easy-to-implement and automatic method that shows some advantage when more than one spuriously correlated feature is present in the data; It outperforms IRM in most settings, hence a promising direction to explore for actually fulfilling the yet unfulfilled promise of IRM.

Weaknesses: The paper lacks (1) a discussion on the key assumptions under which BLOOD works, (2) a discussion on when BLOOD fails to work, and (3) a discussion on the potential source of biases in real data applications.

Given the above strengths and weaknesses, I evaluate the paper as "the contributions are only marginally significant or novel" in the empirical aspect. There is no novel contribution in the technical aspect. Taken together, I would evaluate the paper as "marginally below the acceptance threshold". But I do think there is room for improvement regarding the main weaknesses in my opinion.

---

> ### Author Response · Authors · 2021-11-23
> **Response to Reviewer 9XTF**
>
> Thanks for your feedback. First, we fixed our grammatical errors and language issues.
>
> > For instance, the authors claimed, "... spurious correlations are commonly characterized as simple surface patterns". I believe the authors need to provide more evidence or explanation to support such a claim and evaluate their method when such a premise does not hold.
>
> We rely on the empirical evidence that has been shown in [1,2]. In their experiments,  the biases are characterized as simple surface patterns [1], since the biases are easy to be trained than desired correlations [2]. In addition, when the task is complex so that the biases cannot be easily detected (but still easier to be detected than stable correlations), our algorithm will be performed as ERM until the inner-trained shallow models discover biases with high confidence since the prediction confidence of shallow models will be low at the early stages of training.
>
> > What is the potential source of biases?
>
> In practical scenarios, one cannot easily define all biases since they are unknown. What we can assume is that the spurious correlations are varying across environments, which means any features having covariance with true correlation can be mis-regarded as true correlation.  It is desired that the model does not focus on these spurious correlations, however in most cases the ideal stable correlation is hard to be learned, and many studies are trying to find the stable correlations across the environments. In our study, we gradually remove various spurious correlations so that only stable correlations can remain across all environments.
>
> > Just as in the synthetic experiments, the authors can easily use what BLOOD learned and see if there is any feature whose correlation with the prediction accuracy decreases as the algorithm updates, and also check if those features make any sense.
>
>  In a practical setting, there is no real-world dataset that includes exact bias information or information for confounding high-level features. However,  we believe our experimental results on challenging Camelyon17-wilds and FMoW-wilds can represent empirical evidence that the BLOOD can be expanded to real-world applications because the OOD environments defined in the WILDS are very close to the realistic condition of the dataset. However, for further verification of the algorithm, we added additional experiments on more complex datasets CelebA with known biases in Appendix B. The results in Figure 8 demonstrate that the shallow models can discover biases differently for each learning step conditioned on the $\theta$ for each step, and the effect of bias feature on the shallow model decreases as the model become unbiased.
>
> [1] Utama, Prasetya Ajie, Nafise Sadat Moosavi, and Iryna Gurevych. "Mind the Trade-off: Debiasing NLU Models without Degrading the In-distribution Performance." Proceedings of the 58th Annual Meeting of the Association for Computational Linguistics. 2020.
>
> [2] Geirhos, Robert, et al. "Shortcut learning in deep neural networks." Nature Machine Intelligence 2.11 (2020): 665-673.

---

### Official Review · Reviewer_r5uG · 2021-10-31

**Correctness:** 2
**Technical Novelty And Significance:** 3
**Empirical Novelty And Significance:** 2
**Recommendation:** 5
**Confidence:** 4

**Main Review:**

Strengths:
+ OOD is challenging and the proposed method is novel and interesting. The dynamic re-grouping (rather than a one-shot grouping) is one of the inventions of the paper.
+ Experimental results are good.

Weaknesses:
- the author emphasizes the necessity of using a shallow model in the LL problem (lines 2-7 in Alg.1). However, there is no ablation study about the shallow model. What can happen if the model is deep and does not overfit? Also, line 9 shows that the upper-level model uses the same \theta, indicating that the UL model is a shallow model. Why not decouple the two models?
- the formulation/algorithm is not really a bilevel optimization formulation/algorithm. When optimizing the upper-level variable \theta, the gradient should not just depend on the current \theta, but should also differentiate with respect to \theta through the lower-level optimization variables (\phi_i, the grouping variables, and the q_j). The algorithm is more similar to an alternative optimization of several sets of variables. See [1,2,3]. The author(s) did not cite papers on bi-level optimization.
- the methods mostly rely on existing work. For example, group DRO has been proposed in [4]. The new dynamic regrouping does bring more improvement in OOD accuracy, so this is just a minor weakness.
- In the experiments, there are several correlations not defined clearly, thus affecting the interpretation of the results.
- there is a lack of ablation studies: what if the bias identification is not accurate? It seems that the success will depend on the identification of the bias.
- typos: line 8 of Alg. 1 the denominator seems incorrect; line 9 of Alg. 1, the second q_j should have a negative sign before it.

**Summary Of The Paper:**

The paper studies the problem of OOD classification: the test data and training data distribution can have different spurious feature-class dependencies. The goal is to have high test accuracy of a model trained in one environment. The paper designs a bi-level optimization problem: the lower-level is to re-group data in an environment (distribution) into two groups (correct and incorrect predictions), and the upper-level is to train a model that has different weights over the two groups to focus on the examples that have the changed spurious feature-class dependencies. Experiments on two real-world datasets and synthetic datasets show significantly improved OOD accuracy. There is no theoretical analysis of the method.

**Summary Of The Review:**

The studied problem is important and the empirical results are good. The major weaknesses are the proposed algorithm formulation, motivation, and the lack of ablation studies to verify the need/motivation of the components of the algorithm. There is no formal analysis or explanation -- most explanations are ad hoc.

---

> ### Author Response · Authors · 2021-11-23
> **Response to Reviewer r5uG**
>
> > the author emphasizes the necessity of using a shallow model in the LL problem (lines 2-7 in Alg.1). However, there is no ablation study about the shallow model. What can happen if the model is deep and does not overfit? Also, line 9 shows that the upper-level model uses the same \theta, indicating that the UL model is a shallow model.
>
> > there is a lack of ablation studies: what if the bias identification is not accurate? It seems that the success will depend on the identification of the bias.
>
> First, shallow models in our algorithm are already deep since they are just inner optimized on each training environment from $\theta$. Although our model itself is not shallow enough compared to the conventional definition of a shallow model, our model can be still regarded as a shallow model after a few shots of training in the inner loop as shown in the experiments in Appendix B. In Figure 6, our shallow model effectively learns biases throughout the training process.
>
> For the case, when shallow models do not overfit to bias, our algorithm still can minimize the effect of bias, because our framework performs re-grouping of environments and applies group-based worst-case risk optimization. When the shallow model did not learn bias, the temporarily trained shallow model re-groups the environments and interpolates these newly generated groups while optimizing with groupDRO. As proved in [5], even when unstable features are all ignored by a trained classifier (it is one of its degenerate cases), regrouping still obtains the oracle distribution by directly interpolating distributions of new groups while optimizing over worst-case risk. Even if a model is not trained enough to represent any correlations that exist in the dataset, at least, it can be keep trained until it learns any correlations with  ERM. Therefore, our framework is designed to guarantee the OOD performance even when a shallow model fails to learn biases.
>
> > Why not decouple the two models?
>
> One of the problems of previous studies is they adopt a separate network as their shallow model or weak learner and once it is trained then it remains static throughout the entire training process. In this type of approach, once trained, a shallow model is limited in representing diverse biases in the dataset. Only biases in major effects could be learned one time and de-biased by following methods. Furthermore, because a separate robust model is trained with a fixed shallow model, as the training progresses, how much the bias information from the shallow model should be applied to the robust model is not clear. For example, [1] pointed out the problem that the effective training data size for the main model is reduced because of continuous down-weighting of training samples. For this problem, they propose an annealing mechanism which flattens the final probability to enable the model to gradually learn from all training examples.
> However, in our framework, we address the above two problems at the same time by integrating the role of a shallow model with a robust main model based on a bi-level optimization scheme. We temporarily produce a shallow model in the inner loop of the bi-level scheme with a few-shot training. However, these shallow models are initialized with the main parameters $\theta$, therefore, as the training process progresses, the effect of ‘the learned and debiased biases’ smoothly decreases. This effect is well represented in the confidence graph, figure.7, in Appendix B. It shows the confidence trend of a shallow model on its incorrect predictions. Although a shallow model keeps learning biases with few-shot learning, its confidence gradually decreases as the underlying robust model becomes more robust. With this unified bi-level learning approach, we not only can de-bise diverse biases but also stabilize the training process of a robust model.
>
> > the formulation/algorithm is not really a bilevel optimization formulation/algorithm. When optimizing the upper-level variable \theta, the gradient should not just depend on the current \theta, but should also differentiate with respect to \theta through the lower-level optimization variables (\phi_i, the grouping variables, and the q_j). The algorithm is more similar to an alternative optimization of several sets of variables. See [1,2,3]. The author(s) did not cite papers on bi-level optimization.
>
> We cannot find the paper titles [1, 2, 3, 4] from your comments.
> The formulation of our algorithm is a bi-level learning setup since the inner optimization approximately solves the inner objective with one or multiple steps of stochastic gradient descent. This formulation is similar to meta-learning algorithms with bi-level learning objectives [2, 3].

---

> ### Author Response · Authors · 2021-11-23
> **Response to Reviewer r5uG #2**
>
> > the methods mostly rely on existing work. For example, group DRO has been proposed in [4]. The new dynamic regrouping does bring more improvement in OOD accuracy, so this is just a minor weakness.
>
>  GroupDRO can guarantee worst-case risk performance over human-defined groups, but how one can define optimal groups for OOD performance is not clear. Only to address this issue, several works have been proposed. For example, [4] and [5] proposed training methods that try to address grouping problems without any prior bias information. Those methods alleviate human efforts for identifying biases by data grouping and provide multi-stage training procedures for OOD generalization. In the same context, our BLOOD also aims to provide a novel framework that can provide group robustness without any training group information or prior bias knowledge. As in other related works, it is an important problem with significant contribution in OOD generalization methodology. Currently, there is no other OOD training method that can be applied in an end-to-end manner and does not require a separate network or retraining of the model.
>
> > In the experiments, there are several correlations not defined clearly, thus affecting the interpretation of the results.
>
> The correlation in the paper mostly indicates a statistical relationship between high-level features in which the model exploits for prediction and the target variable. In the experiments, we measure such correlations as Pearson correlation. As your feedback, we defined correlations in our experiments.
>
> > typos: line 8 of Alg. 1 the denominator seems incorrect; line 9 of Alg. 1, the second q_j should have a negative sign before it.
>
> Thank for the correction. We modified line 9 of Algorithm 1. But the sign of the $q_j$ is correct.
>
> [1] Utama, Prasetya Ajie, Nafise Sadat Moosavi, and Iryna Gurevych. "Towards Debiasing NLU Models from Unknown Biases." Proceedings of the 2020 Conference on Empirical Methods in Natural Language Processing (EMNLP). 2020.
>
> [2] Finn, Chelsea, Pieter Abbeel, and Sergey Levine. "Model-agnostic meta-learning for fast adaptation of deep networks." International Conference on Machine Learning. PMLR, 2017.
>
> [3] Rajeswaran, Aravind, et al. "Meta-Learning with Implicit Gradients." Advances in Neural Information Processing Systems 32 (2019): 113-124.
>
> [4] Liu, Evan Z., et al. "Just train twice: Improving group robustness without training group information." International Conference on Machine Learning. PMLR, 2021.
>
> [5] Yujia Bao, Shiyu Chang, Regina Barzilay. "Predict then Interpolate: A Simple Algorithm to Learn Stable Classifiers." Proceedings of the 38th International Conference on Machine Learning, PMLR 139:640-650, 2021.

---

### Official Review · Reviewer_t2w7 · 2021-11-02

**Correctness:** 2
**Technical Novelty And Significance:** 1
**Empirical Novelty And Significance:** 2
**Recommendation:** 3
**Confidence:** 3

**Main Review:**


-------------------------- pros
1. This paper is overall well written.
2. The proposed method achieves promising performance empirically.

-------------------------- cons
1. The technical novelty is limited since bilevel optimization framework has been widely used in machine learning.
2. Precisely, the formulation of the proposed method is not bilevel optimization framework but triple-level like optimization framework since in the outer-loop it is a min-max optimization problem. Authors should make it more clear.
3. This paper lacks theoretical supports including the perspectives of the generalization and optimization convergence. As mentioned in 2, the optimization problem is different from ordinary bilevel optimization problem.
4. Empirically, is the proposed method more efficient than other baselines? Besides, more discussions should be added to explain why it works. Further, experiments might be conducted on more real-world datasets to support the claims.

------------------------- minor comments
1. In the abstract, what is DRO?
2. In the synthetic experiments, what is the correlation? Pearson (linear) correlation?

**Summary Of The Paper:**

This paper considers the out-of-distribution (OOD) problem. To automatically remove multiple unknown types of biases without prior information, this paper proposes a bi-level learning framework, where in the inner-loop it aims to detect the biases while aiming to remove them in the outer-loop via group Distributionally Robust Optimization (DRO). Experimental results on synthetic and real datasets illustrates the effectiveness of the proposed method over other baselines.

**Summary Of The Review:**

Overall, I vote for rejecting mainly because I think the technical novelty is limited and it lacks theoretical supports.

---

> ### Author Response · Authors · 2021-11-23
> **Response to Reviewer t2w7**
>
> > The technical novelty is limited since bilevel optimization framework has been widely used in machine learning.
>
> A bi-level learning approach is a widely adopted method, not because it is a unique and novel methodology, but because we can utilize its bi-level nature for simplifying the problem or able to provide a more efficient solution with such formulation. We are not arguing that the bi-level approach itself is an unprecedented novelty, but we nicely formulated that pre-existing OOD problem appropriates for the bi-level learning paradigm, so that it can have more efficient OOD performance and easily adopted model-agnostically, as we mentioned in the introduction. Most other OOD methods are intended for specific application domains where human-annotated biases are available. Our framework automatically figures out biases existing in the dataset without any human help and leverages the worst-case risk-based optimization method (group DRO) for optimal OOD generalization performance. Although group DRO provides theoretical performance guarantee for worst-case risk among given groups, their assumption on data groups is pretty much heuristic and needs human experts’ prior knowledge on the data. Therefore, its performance largely depends on how groups are constructed. For this reason, recently, several methods for the grouping of training data grouping have been proposed  [1, 2]. As shown in those researches, providing optimal groups for OOD training is not only important but also difficult problem, especially for practical datasets, which have no prior knowledge on biases. Furthermore, they adopted one-time regrouping of environments that can not reflect diverse biases that exist in the dataset. In contrast, our method applies dynamic regrouping of the environment per each batch data while training a model. This repetitive regrouping scheme enables our method to reflect diverse biases in the dataset to diverse grouping. This dynamic regrouping allows a model to automatically de-bias multiple unknown biases in the given dataset.
>
> > Precisely, the formulation of the proposed method is not bilevel optimization framework but triple-level like optimization framework since in the outer-loop it is a min-max optimization problem. Authors should make it more clear.
>
> Our algorithm contains min-max optimization problem in the outer-loop, however, the group weights $q$ are not connected with $\theta$. Thus, the optimization of $\theta$ in the outer-loop is nested with only the inner optimization process.
>
> > This paper lacks theoretical supports including the perspectives of the generalization and optimization convergence. As mentioned in 2, the optimization problem is different from ordinary bilevel optimization problem.
>
> Since our method is theoretically based on the online group DRO and PI, we did not specifically show the theoretical guarantee for convergence and generalization. For unknown bias discovery, our method leverages both shallow model-based approach and grouping-based method at the same time. Learning from mistakes of a shallow model(or weak learner) for bias discovery is a very effective approach with empirical benefits on many tasks, however, the extent to which its assumption holds is not guaranteed yet. As an alternative, grouping-based approach, such as [1],  apply re-grouping of given environments based on each environment-specific classifier. This approach interpolates an oracle distribution of data by optimizing over re-grouped environments, and also provides a theoretical guarantee with conditional probability and covariance. However, it does not specifically consider the effect of diverse biases when re-grouping the data, and performs grouping only once.
> Our method combines the advantages of the above two approaches. We adopt a shallow model assumption for the ease of bias discovery, but also perform re-grouping of given environments to provide the upper bound of worst-case risk optimization. In this way, easily learnable biases are discovered by shallow model and, even if some biases are not discovered, their effects can be minimized by re-grouping of environments and group-based optimization(groupDRO) method. We did not show the formal convergence proof of our bi-level learning algorithm, because its convergence is the mere extension of online groupDRO convergence proof. Although our algorithm is theoretically supported by both PI and groupDRO research,  we are going to provide more analysis in the future.

---

> ### Author Response · Authors · 2021-11-23
> **Response to Reviewer t2w7 #2**
>
> > Empirically, is the proposed method more efficient than other baselines? Besides, more discussions should be added to explain why it works. Further, experiments might be conducted on more real-world datasets to support the claims.
>
> We provided synthetical analyses of BLOOD with Colored MNIST because the real-world datasets do not provide exact bias information for training. We updated additional evaluation results and discussion on a more complex and practical dataset (CelebA) dataset which has approximately known bias information, see Appendix B. In real practical applications, the bias information is usually not available, and under such conditions,  the performance of other OOD methods, such as group DRO is severely degraded. However, our algorithm is designed to keep discovering unknown biases and dynamically re-group the environments for de-biasing a model for every training step. The evaluation result of our algorithm on practical datasets, Camelyon17-wilds and FMoW-wilds, shows the superiority of our framework compared to other types of OOD approaches, such as IRM, which is theoretically sound but shows weak OOD generalization performance on challenging OOD test environments.
>
> **minor comments**
> > In the abstract, what is DRO?
>
> Thank you for your feedback. We modified our abstract as you pointed.
>
> > In the synthetic experiments, what is the correlation? Pearson (linear) correlation?
>
> The correlation in the paper mostly indicates a statistical relationship between high-level features, in which the model exploits for prediction and the target variable. In the experiments, these correlations are measured as Pearson correlation. We found that we missed ‘Pearson’ in the description of Figure 3 in our paper. Thanks for your correction.
>
> [1]  Yujia Bao, Shiyu Chang, Regina Barzilay. "Predict then Interpolate: A Simple Algorithm to Learn Stable Classifiers." Proceedings of the 38th International Conference on Machine Learning, PMLR 139:640-650, 2021.
>
> [2] Liu, Evan Z., et al. "Just train twice: Improving group robustness without training group information." International Conference on Machine Learning. PMLR, 2021.

---

### Official Review · Reviewer_gQqn · 2021-11-04

**Correctness:** 3
**Technical Novelty And Significance:** 3
**Empirical Novelty And Significance:** 3
**Recommendation:** 5
**Confidence:** 4

**Main Review:**

The paper proposes a novel approach for domain generalization. Overall, the paper is clearly written. However, it gets hard to follow in certain sections but it doesn't obscure the overall understanding (please see below).

Major concerns:
- This paper is primarily empirical. While the results on the three datasets included in the experiments look promising, comparison on other datasets can strengthen the empirical efficacy of the proposed method BLOOD. For example, it would be interesting to evaluate and compare with datasets in DomainBed https://github.com/facebookresearch/DomainBed/. I encourage authors to compare their method with ERM and the best alternatives in [1]. This is my major concern.

- Alternatively, it would also be interesting to include more details on the diversity of biases in the groups obtained with different shallow learners at least on the toy MNIST dataset.

- The choice of shallow models used for bi-level learning is not clear in the experimental section.

Other bits:
- It would help a reader if a few lines can explain the notation used in Algorithm 1. Maybe add some notation in a paragraph before Section 3. For example, the following things were unclear in the first past (I found the description as I read Section 3 but it would be better to define the notation upfront): (i) the notation with (.) and (x); (ii) The role of q is not clear.
- Why only worst-case region accuracy is included for FMoW.
- How are hyperparameters tuned for Algorithm 1?

[1] Gulrajani and Lopez-Paz, In Search of Lost Domain Generalization. https://openreview.net/pdf?id=lQdXeXDoWtI


**Summary Of The Paper:**

The paper proposes a new method called BLOOD for the problem of domain generalization. The proposed procedure has two broad steps: (i) Identify groups with different biases (ii) Minimize risk on each of these groups with the group DRO idea. Authors show promising results on colored MNIST and Camelyon17-wilds, FMoW-wilds.

**Summary Of The Review:**

Overall, the paper is clear and proposes a novel method. However, the empirical evaluation can benefit from more exhaustive comparison, for example on Domain Bed. It is also not clear if the motivation to use shallow learners in the inner loop of bi-level optimization is practically achieved even in the toy set. Analysis highlighting the characteristics of the biases identified by groups can be insightful.

---

> ### Author Response · Authors · 2021-11-23
> **Response to Reviewer gQqn**
>
> **Major concerns:**
> > This paper is primarily empirical. While the results on the three datasets included in the experiments look promising, comparison on other datasets can strengthen the empirical efficacy of the proposed method BLOOD. For example, it would be interesting to evaluate and compare with datasets in DomainBed https://github.com/facebookresearch/DomainBed/.
>
> We thank to the reviewer for recommending a useful OOD dataset, DomainBed. It would be very helpful for showing the strength of our framework. We are going to evaluate our model on DomainBed and will update results, and in our experiment section, there already exists the evaluation result of some part of the DomainBed dataset (Camelyon17-wilds and FMoW-wilds). Furthermore, we believe Camelyon17-wilds and FMoW-wilds are very realistic OOD datasets that can effectively demonstrate the practical OOD performance of the model on real-world applications.
>
> > Alternatively, it would also be interesting to include more details on the diversity of biases in the groups obtained with different shallow learners at least on the toy MNIST dataset.
>
> We added our new experimental results in Appendix B. The evaluation results show the ability of the shallow models to re-discover biases and their confidence in them. It also shows how the shallow models’ predictions change according to the learning steps. Note that the shallow models are trained based on the robust parameters $\theta$. The results demonstrate that our shallow models can keep discovering various biases with different correlations for each learning step, even when conditioned on the $\theta$ for each step. The effect of biases in the groups at each training step keeps changing because we are iterating de-biasing and bias discovering procedures while training a model. Furthermore, because we are training environment-specific shallow models for regrouping the environment in the batch, even when the shallow models did not learn unstable correlations, the optimization with group DRO can guarantee worst-case risk upper bound with oracle distribution, as stated in ‘predict and interpolate’ paper.
>
> > The choice of shallow models used for bi-level learning is not clear in the experimental section.
>
> In the paper, such as [1], a shallow model is obtained by freezing the model in the early stages of learning. In contrast, our algorithm doesn’t explicitly choose a single fixed shallow model; instead, it dynamically obtains them by temporarily training a model based on $\theta$ with a few steps of gradient descent in each batch. The parameters of the shallow model $\phi_i$ are obtained after the few-shot learning of the data sampled from specific-environment $e_i$. Because a shallow model is obtained for each environment and used for re-grouping the other environment, if there are two environments, then we have two shallow models, and each shallow model regroups the other environment which it is not trained on. Therefore, we can obtain four new groups for group-based OOD optimization.
>
> **Other bits:**
> > It would help a reader if a few lines can explain the notation used in Algorithm 1. Maybe add some notation in a paragraph before Section 3. For example, the following things were unclear in the first past (I found the description as I read Section 3 but it would be better to define the notation upfront): (i) the notation with (.) and (x); (ii) The role of q is not clear.
>
> As your comment, we found that we missed the explanation of $\odot$ and $\otimes$ of group weights $q$. We updated our paper, added the explanations of notations of $q$.
>
> > Why only worst-case region accuracy is included for FMoW.
>
> In the FMoW-wilds dataset, there are two kinds of shifts, first, the time of picture have taken, which is temporal domain shift, and sub-population shift which means the different regions have different numbers of data samples (Africa includes much fewer images). Therefore, the worst-case region accuracy is appropriate for representing both a domain generalization over time as well as subpopulation shift over regions.
>
> > How are hyperparameters tuned for Algorithm 1?
>
> We did an exhaustive grid search for tuning the hyperparameters of algorithm 1.
>
> [1] Utama, Prasetya Ajie, Nafise Sadat Moosavi, and Iryna Gurevych. "Towards Debiasing NLU Models from Unknown Biases." Proceedings of the 2020 Conference on Empirical Methods in Natural Language Processing (EMNLP). 2020.

---

> > ### Comment · Reviewer_gQqn · 2021-11-25
> > **How do you do grid search?**
> >
> > Just a minor question from your responses, how do you do grid search, i.e., what is the value that you optimize for in your grid search? Is it accuracy on ID data or OOD data?

---

> > > ### Author Response · Authors · 2021-11-26
> > > **Response**
> > >
> > > Thank you for your quick comment.
> > >
> > > Since the Colored MNIST task is not computationally complex, we did grid search over inner learning step $\alpha$, outer learning step $\beta$, and group weights $\gamma$.
> > > We considered $\alpha$, $\beta$ $\in$ {0.1, 0.01, 0.001, 0.0001} and $\gamma$ $\in$ { 0.1, 0.01}.
> > >
> > > For real-world datasets, Camelyon17-wilds and FMoW-wilds, we restrict $\alpha=\beta$ to reduce the computational cost for searching.
> > > For $\gamma$, we used an official hyperparameter for group DRO on WILDS, 0.01.
> > > We followed official hyperparameters for FMoW-wilds dataset, however, we found that the official learning rate for Camelyon17-wilds is too large in our case. Thus, we did grid search to find better hyperparameters.
> > > We considered official hyperparameters and set the searching range as {$\times 1/100$, $\times 1/10$,  $\times 1$}.
> > >
> > > > Is it accuracy on ID data or OOD data?
> > >
> > > We tuned our hyperparameters on IID validation data for Colored MNIST and OOD validation data for WILDS datasets.
> > > For WILDS, the official results are tuned on the OOD validation set, we followed their method for tuning hyperparameters.

---

### Decision · Program_Chairs · 2022-01-20

**Decision:**

Reject

**Comment:**

The paper studies the problem of OOD classification: the test data and training data distribution can have different spurious feature-class dependencies.

The reviewers have stated that the proposed procedure is a natural choice, with simple implementation. Another positive point is that it could easily be incorporated in many off-the-shelf machine learning training algorithms.

Yet, the technical novelty was mentioned to be limited. The bilevel optimization point of view and the connection with min-max optimization problems raised some concerns, as the vocabulary used could be misleading.
It was also raised that the paper lacks theoretical supports: no formal analysis, most explanations are ad hoc, etc.